# Generating Videos with Scene Dynamics

**Carl Vondrick**
MIT
vondrick@mit.edu

**Hamed Pirsiavash**
UMBC
hpirsiav@umbc.edu

**Antonio Torralba**
MIT
torralba@mit.edu

## Abstract

We capitalize on large amounts of unlabeled video in order to learn a model of scene dynamics for both video recognition tasks (e.g. action classification) and video generation tasks (e.g. future prediction). We propose a generative adversarial network for video with a spatio-temporal convolutional architecture that untangles the scene's foreground from the background. Experiments suggest this model can generate tiny videos up to a second at full frame rate better than simple baselines, and we show its utility at predicting plausible futures of static images. Moreover, experiments and visualizations show the model internally learns useful features for recognizing actions with minimal supervision, suggesting scene dynamics are a promising signal for representation learning. We believe generative video models can impact many applications in video understanding and simulation.

## 1   Introduction

Understanding object motions and scene dynamics is a core problem in computer vision. For both video recognition tasks (e.g., action classification) and video generation tasks (e.g., future prediction), a model of how scenes transform is needed. However, creating a model of dynamics is challenging because there is a vast number of ways that objects and scenes can change.

In this work, we are interested in the fundamental problem of learning how scenes transform with time. We believe investigating this question may yield insight into the design of predictive models for computer vision. However, since annotating this knowledge is both expensive and ambiguous, we instead seek to learn it directly from large amounts of in-the-wild, unlabeled video. Unlabeled video has the advantage that it can be economically acquired at massive scales yet contains rich temporal signals "for free" because frames are temporally coherent.

With the goal of capturing some of the temporal knowledge contained in large amounts of unlabeled video, we present an approach that learns to generate tiny videos which have fairly realistic dynamics and motions. To do this, we capitalize on recent advances in generative adversarial networks [9, 31, 4], which we extend to video. We introduce a two-stream generative model that explicitly models the foreground separately from the background, which allows us to enforce that the background is stationary, helping the network to learn which objects move and which do not.

Our experiments suggest that our model has started to learn about dynamics. In our generation experiments, we show that our model can generate scenes with plausible motions.[1] We conducted a psychophysical study where we asked over a hundred people to compare generated videos, and people preferred videos from our full model more often. Furthermore, by making the model conditional on an input image, our model can sometimes predict a plausible (but "incorrect") future. In our recognition experiments, we show how our model has learned, without supervision, useful features for human action classification. Moreover, visualizations of the learned representation suggest future generation may be a promising supervisory signal for learning to recognize objects of motion.

The primary contribution of this paper is showing how to leverage large amounts of unlabeled video in order to acquire priors about scene dynamics. The secondary contribution is the development of a generative model for video. The remainder of this paper describes these contributions in detail. In section 2, we describe our generative model for video. In section 3, we present several experiments to analyze the generative model. We believe that generative video models can impact many applications, such as in simulations, forecasting, and representation learning.

## 1.1 Related Work

This paper builds upon early work in generative video models [29]. However, previous work has focused mostly on small patches, and evaluated it for video clustering. Here, we develop a generative video model for natural scenes using state-of-the-art adversarial learning methods [9, 31]. Conceptually, our work is related to studies into fundamental roles of time in computer vision [30, 12, 2, 7, 24]. However, here we are interested in generating short videos with realistic temporal semantics, rather than detecting or retrieving them.

Our technical approach builds on recent work in generative adversarial networks for image modeling [9, 31, 4, 47, 28], which we extend to video. To our knowledge, there has been relatively little work extensively studying generative adversarial networks for video. Most notably, [22] also uses adversarial networks for video frame prediction. Our framework can generate videos for longer time scales and learn representations of video using unlabeled data. Our work is also related to efforts to predict the future in video [33, 22, 43, 50, 42, 17, 8, 54] as well as concurrent work in future generation [6, 15, 20, 49, 55]. Often these works may be viewed as a generative model conditioned on the past frames. Our work complements these efforts in two ways. Firstly, we explore how to generate videos from scratch (not conditioned on the past). Secondly, while prior work has used generative models in video settings mostly on a single frame, we jointly generate a sequence of frames (32 frames) using spatio-temporal convolutional networks, which may help prevent drifts due to errors accumulating.

We leverage approaches for recognizing actions in video with deep networks, but apply them for video generation instead. We use spatio-temporal 3D convolutions to model videos [40], but we use fractionally strided convolutions [51] instead because we are interested in generation. We also use two-streams to model video [34], but apply them for video generation instead of action recognition. However, our approach does not explicitly use optical flow; instead, we expect the network to learn motion features on its own. Finally, this paper is related to a growing body of work that capitalizes on large amounts of unlabeled video for visual recognition tasks [18, 46, 37, 13, 24, 25, 3, 32, 26, 27, 19, 41, 42, 1]. We instead leverage large amounts of unlabeled video for generation.

## 2 Generative Models for Video

In this section, we present a generative model for videos. We propose to use generative adversarial networks [9], which have been shown to have good performance on image generation [31, 4].

### 2.1 Review: Generative Adversarial Networks

The main idea behind generative adversarial networks [9] is to train two networks: a generator network $G$ tries to produce a video, and a discriminator network $D$ tries to distinguish between "real" videos and "fake" generated videos. One can train these networks against each other in a min-max game where the generator seeks to maximally fool the discriminator while simultaneously the discriminator seeks to detect which examples are fake:

$$\min_{w_G} \max_{w_D} \mathbb{E}_{x \sim p_x(x)} \left[ \log D(x; w_D) \right] + \mathbb{E}_{z \sim p_z(z)} \left[ \log \left( 1 - D(G(z; w_G); w_D) \right) \right] \quad (1)$$

where $z$ is a latent "code" that is often sampled from a simple distribution (such as a normal distribution) and $x \sim p_x(x)$ samples from the data distribution. In practice, since we do not know the true distribution of data $p_x(x)$, we can estimate the expectation by drawing from our dataset.

Since we will optimize Equation 1 with gradient based methods (SGD), the two networks $G$ and $D$ can take on any form appropriate for the task as long as they are differentiable with respect to parameters $w_G$ and $w_D$. We design a $G$ and $D$ for video.

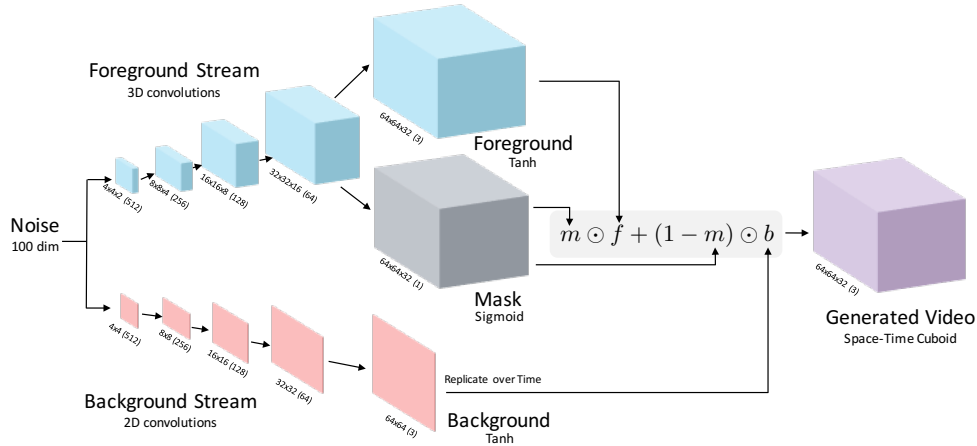

Figure 1: **Video Generator Network:** We illustrate our network architecture for the generator. The input is 100 dimensional (Gaussian noise). There are two independent streams: a moving foreground pathway of fractionally-strided spatio-temporal convolutions, and a static background pathway of fractionally-strided spatial convolutions, both of which up-sample. These two pathways are combined to create the generated video using a mask from the motion pathway. Below each volume is its size and the number of channels in parenthesis.

## 2.2 Generator Network

The input to the generator network is a low-dimensional latent code $z \in \mathbb{R}^d$. In most cases, this code can be sampled from a distribution (e.g., Gaussian). Given a code $z$, we wish to produce a video.

We design the architecture of the generator network with a few principles in mind. Firstly, we want the network to be invariant to translations in both space and time. Secondly, we want a low-dimensional $z$ to be able to produce a high-dimensional output (video). Thirdly, we want to assume a stationary camera and take advantage of the the property that usually only objects move. We are interested in modeling object motion, and not the motion of cameras. Moreover, since modeling that the background is stationary is important in video recognition tasks [44], it may be helpful in video generation as well. We explore two different network architectures:

**One Stream Architecture:** We combine spatio-temporal convolutions [14, 40] with fractionally strided convolutions [51, 31] to generate video. Three dimensional convolutions provide spatial and temporal invariance, while fractionally strided convolutions can upsample efficiently in a deep network, allowing $z$ to be low-dimensional. We use an architecture inspired by [31], except extended in time. We use a five layer network of $4 \times 4 \times 4$ convolutions with a stride of 2, except for the first layer which uses $2 \times 4 \times 4$ convolutions (time $\times$ width $\times$ height). We found that these kernel sizes provided an appropriate balance between training speed and quality of generations.

**Two Stream Architecture:** The one stream architecture does not model that the world is stationary and usually only objects move. We experimented with making this behavior explicit in the model. We use an architecture that enforces a static background and moving foreground. We use a two-stream architecture where the generator is governed by the combination:

$$G_2(z) = m(z) \odot f(z) + (1 - m(z)) \odot b(z). \qquad (2)$$

Our intention is that $0 \geq m(z) \geq 1$ can be viewed as a spatio-temporal mask that selects either the foreground $f(z)$ model or the background model $b(z)$ for each pixel location and timestep. To enforce a background model in the generations, $b(z)$ produces a spatial image that is replicated over time, while $f(z)$ produces a spatio-temporal cuboid masked by $m(z)$. By summing the foreground model with the background model, we can obtain the final generation. Note that $\odot$ is element-wise multiplication, and we replicate singleton dimensions to match its corresponding tensor. During learning, we also add to the objective a small sparsity prior on the mask $\lambda \|m(z)\|_1$ for $\lambda = 0.1$, which we found helps encourage the network to use the background stream.

We use fractionally strided convolutional networks for $m(z)$, $f(z)$, and $b(z)$. For $f(z)$, we use the same network as the one-stream architecture, and for $b(z)$ we use a similar generator architecture to [31]. We only use their architecture; we do not initialize with their learned weights. To create the mask $m(z)$, we use a network that shares weights with $f(z)$ except the last layer, which has only one output channel. We use a sigmoid activation function for the mask. We visualize the two-stream architecture in Figure 1. In our experiments, the generator produces $64 \times 64$ videos for 32 frames, which is a little over a second.

### 2.3 Discriminator Network

The discriminator needs to be able to solve two problems: firstly, it must be able to classify realistic scenes from synthetically generated scenes, and secondly, it must be able to recognize realistic motion between frames. We chose to design the discriminator to be able to solve both of these tasks with the same model. We use a five-layer spatio-temporal convolutional network with kernels $4 \times 4 \times 4$ so that the hidden layers can learn both visual models and motion models. We design the architecture to be reverse of the foreground stream in the generator, replacing fractionally strided convolutions with strided convolutions (to down-sample instead of up-sample), and replacing the last layer to output a binary classification (real or not).

### 2.4 Learning and Implementation

We train the generator and discriminator with stochastic gradient descent. We alternate between maximizing the loss w.r.t. $w_D$ and minimizing the loss w.r.t. $w_G$ until a fixed number of iterations. All networks are trained from scratch. Our implementation is based off a modified version of [31] in Torch7. We used a more numerically stable implementation of cross entropy loss to prevent overflow. We use the Adam [16] optimizer and a fixed learning rate of $0.0002$ and momentum term of $0.5$. The latent code has $100$ dimensions, which we sample from a normal distribution. We use a batch size of $64$. We initialize all weights with zero mean Gaussian noise with standard deviation $0.01$. We normalize all videos to be in the range $[-1, 1]$. We use batch normalization [11] followed by the ReLU activation functions after every layer in the generator, except the output layers, which uses tanh. Following [31], we also use batch normalization in the discriminator except for the first layer and we instead use leaky ReLU [48]. Training typically took several days on a GPU.

## 3 Experiments

We experiment with the generative adversarial network for video (VGAN) on both generation and recognition tasks. We also show several qualitative examples online.

### 3.1 Unlabeled Video Dataset

We use a large amount of unlabeled video to train our model. We downloaded over two million videos from Flickr [39] by querying for popular Flickr tags as well as querying for common English words. From this pool, we created two datasets:

**Unfiltered Unlabeled Videos:** We use these videos directly, without any filtering, for representation learning. The dataset is over $5,000$ hours. **Filtered Unlabeled Videos:** To evaluate generations, we use the Places2 pre-trained model [53] to automatically filter the videos by scene category. Since image/video generation is a challenging problem, we assembled this dataset to better diagnose strengths and weaknesses of approaches. We experimented with four scene categories: golf course, hospital rooms (babies), beaches, and train station.

**Stabilization:** As we are interested in the movement of objects and not camera shake, we stabilize the camera motion for both datasets. We extract SIFT keypoints [21], use RANSAC to estimate a homography (rotation, translation, scale) between adjacent frames, and warp frames to minimize background motion. When the homography moved out of the frame, we fill in the missing values using the previous frames. If the homography has too large of a re-projection error, we ignore that segment of the video for training, which only happened 3% of the time. The only other pre-processing we do is normalizing the videos to be in the range $[-1, 1]$. We extract frames at native frame rate (25 fps). We use 32-frame videos of spatial resolution $64 \times 64$.

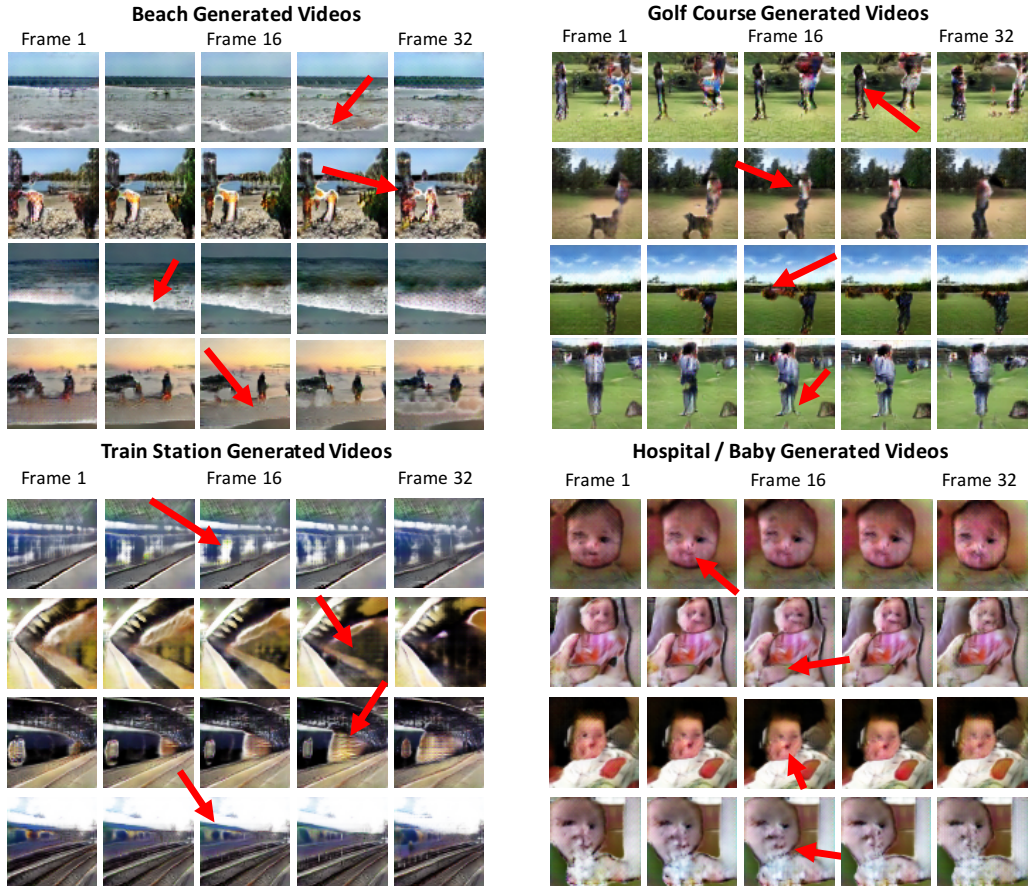

Figure 2: **Video Generations:** We show some generations from the two-stream model. The red arrows highlight motions. Please see `http://mit.edu/vondrick/tinyvideo` for animated movies.

## 3.2  Video Generation

We evaluate both the one-stream and two-stream generator. We trained a generator for each scene category in our filtered dataset. We perform both a qualitative evaluation as well as a quantitative psychophysical evaluation to measure the perceptual quality of the generated videos.

**Qualitative Results:** We show several examples of the videos generated from our model in Figure 2. We observe that a) the generated scenes tend to be fairly sharp and that b) the motion patterns are generally correct for their respective scene. For example, the beach model tends to produce beaches with crashing waves, the golf model produces people walking on grass, and the train station generations usually show train tracks and a train with windows rapidly moving along it. While the model usually learns to put motion on the right objects, one common failure mode is that the objects lack resolution. For example, the people in the beaches and golf courses are often blobs. Nevertheless, we believe it is promising that our model can generate short motions. We visualize the behavior of the two-stream architecture in Figure 3.

**Baseline:** Since to our knowledge there are no existing large-scale generative models of video ([33] requires an input frame), we develop a simple but reasonable baseline for this task. We train an autoencoder over our data. The encoder is similar to the discriminator network (except producing 100 dimensional code), while the decoder follows the two-stream generator network. Hence, the baseline autoencoder network has a similar number of parameters as our full approach. We then feed examples through the encoder and fit a Gaussian Mixture Model (GMM) with 256 components over the 100 dimensional hidden space. To generate a novel video, we sample from this GMM, and feed the sample through the decoder.

**Evaluation Metric:** We quantitatively evaluate our generation using a psychophysical two-alternative forced choice with workers on Amazon Mechanical Turk. We show a worker two random videos,

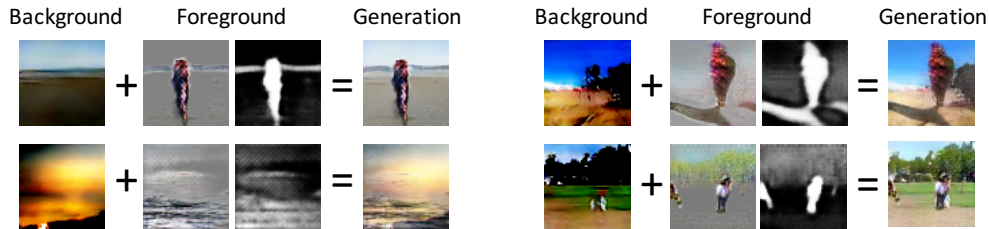

Figure 3: **Streams:** We visualize the background, foreground, and masks for beaches (left) and golf (right). The network generally learns to disentangle the foreground from the background.

| | | Percentage of Trials | | | |
| --- | --- | --- | --- | --- | --- |
| "Which video is more realistic?" | Golf | Beach | Train | Baby | Mean |
| Random Preference | 50 | 50 | 50 | 50 | 50 |
| Prefer VGAN Two Stream over Autoencoder | 88 | 83 | 87 | 71 | 82 |
| Prefer VGAN One Stream over Autoencoder | 85 | 88 | 85 | 73 | 82 |
| Prefer VGAN Two Stream over VGAN One Stream | 55 | 58 | 47 | 52 | 53 |
| Prefer VGAN Two Stream over Real | 21 | 23 | 23 | 6 | 18 |
| Prefer VGAN One Stream over Real | 17 | 21 | 19 | 8 | 16 |
| Prefer Autoencoder over Real | 4 | 2 | 4 | 2 | 3 |

Table 1: **Video Generation Preferences:** We show two videos to workers on Amazon Mechanical Turk, and ask them to choose which video is more realistic. The table shows the percentage of times that workers prefer one generations from one model over another. In all cases, workers tend to prefer video generative adversarial networks over an autoencoder. In most cases, workers show a slight preference for the two-stream model.

and ask them "Which video is more realistic?" We collected over $13,000$ opinions across $150$ unique workers. We paid workers one cent per comparison, and required workers to historically have a 95% approval rating on MTurk. We experimented with removing bad workers that frequently said real videos were not realistic, but the relative rankings did not change. We designed this experiment following advice from [38], which advocates evaluating generative models for the task at hand. In our case, we are interested in perceptual quality of motion. We consider a model X better than model Y if workers prefer generations from X more than generations from Y.

**Quantitative Results:** Table 1 shows the percentage of times that workers preferred generations from one model over another. Workers consistently prefer videos from the generative adversarial network more than an autoencoder. Additionally, workers show a slight preference for the two-stream architecture, especially in scenes where the background is large (e.g., golf course, beach). Although the one-stream architecture is capable of generating stationary backgrounds, it may be difficult to find this solution, motivating a more explicit architecture. The one-stream architecture generally produces high-frequency temporal flickering in the background. To evaluate whether static frames are better than our generations, we also ask workers to choose between our videos and a static frame, and workers only chose the static frame 38% of the time, suggesting our model produces more realistic motion than static frames on average. Finally, while workers generally can distinguish real videos from generated videos, the workers show the most confusion with our two-stream model compared to baselines, suggesting the two-stream generations may be more realistic on average.

### 3.3 Video Representation Learning

We also experimented with using our model as a way to learn unsupervised representations for video. We train our two-stream model with over $5,000$ hours of unfiltered, unlabeled videos from Flickr. We then fine-tune the discriminator on the task of interest (e.g., action recognition) using a relatively small set of labeled video. To do this, we replace the last layer (which is a binary classifier) with a $K$-way softmax classifier. We also add dropout [36] to the penultimate layer to reduce overfitting.

**Action Classification:** We evaluated performance on classifying actions on UCF101 [35]. We report accuracy in Figure 4a. Initializing the network with the weights learned from the generative adversarial network outperforms a randomly initialized network, suggesting that it has learned an useful internal representation for video. Interestingly, while a randomly initialized network under-performs hand-crafted STIP features [35], the network initialized with our model significantly

| Method | Accuracy |
|---|---|
| Chance | 0.9% |
| STIP Features [35] | 43.9% |
| Temporal Coherence [10] | 45.4% |
| Shuffle and Learn [24] | 50.2% |
| VGAN + Random Init | 36.7% |
| VGAN + Logistic Reg | 49.3% |
| **VGAN + Fine Tune** | **52.1%** |
| ImageNet Supervision [45] | 91.4% |

(a) Accuracy with Unsupervised Methods

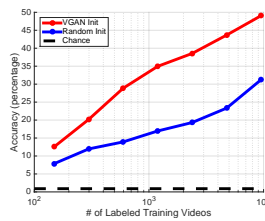

(b) Performance vs # Data

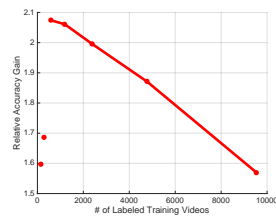

(c) Relative Gain vs # Data

Figure 4: **Video Representation Learning:** We evaluate the representation learned by the discriminator for action classification on UCF101 [35]. **(a)** By fine-tuning the discriminator on a relatively small labeled dataset, we can obtain better performance than random initialization, and better than hand-crafted space-time interest point (STIP) features. Moreover, our model slightly outperforms another unsupervised video representation [24] despite using an order of magnitude fewer learned parameters and only $64 \times 64$ videos. Note unsupervised video representations are still far from models that leverage external supervision. **(b)** Our unsupervised representation with less labeled data outperforms random initialization with all the labeled data. Our results suggest that, with just 1/8th of the labeled data, we can match performance to a randomly initialized network that used all of the labeled data. **(c)** The fine-tuned model has larger relative gain over random initialization in cases with less labeled data. Note that (a) is over all train/test splits of UCF101, while (b,c) is over the first split in order to make experiments less expensive.

outperforms it. We also experimented with training a logistic regression on only the last layer, which performed worse. Finally, our model slightly outperforms another recent unsupervised video representation learning approach [24]. However, our approach uses an order of magnitude fewer parameters, less layers (5 layers vs 8 layers), and low-resolution video.

**Performance vs Data:** We also experimented with varying the amount of labeled training data available to our fine-tuned network. Figure 4b reports performance versus the amount of labeled training data available. As expected, performance increases with more labeled data. The fine-tuned model shows an advantage in low data regimes: even with *one eighth* of the labeled data, the fine-tuned model still beats a randomly initialized network. Moreover, Figure 4c plots the relative accuracy gain over the fine-tuned model and the random initialization (fine-tuned performance divided by random initialized performance). This shows that fine-tuning with our model has larger relative gain over random initialization in cases with less labeled data, showing its utility in low-data regimes.

### 3.4 Future Generation

We investigate whether our approach can be used to generate the future of a static image. Specifically, given a static image $x_0$, can we extrapolate a video of possible consequent frames?

**Encoder:** We utilize the same model as our two-stream model, however we must make one change in order to input the static image instead of the latent code. We can do this by attaching a five-layer convolutional network to the front of the generator which encodes the image into the latent space, similar to a conditional generative adversarial network [23]. The rest of the generator and discriminator networks remain the same. However, we add an additional loss term that minimizes the L1 distance between the input and the first frame of the generated image. We do this so that the generator creates videos consistent with the input image. We train from scratch with the objective:

$$\min_{w_G} \max_{w_D} \mathbb{E}_{x \sim p_x(x)} \left[ \log D(x; w_D) \right] + \mathbb{E}_{x_0 \sim p_{x_0}(x_0)} \left[ \log \left( 1 - D(G(x_0; w_G); w_D) \right) \right]$$
$$+ \mathbb{E}_{x_0 \sim p_{x_0}(x_0)} \left[ \lambda \| x_0 - G^0(x_0; w_G) \|_2^2 \right]$$
(3)

where $x_0$ is the first frame of the input, $G^0(\cdot)$ is the first frame of the generated video, and $\lambda \in \mathcal{R}$ is a hyperparameter. The discriminator will try to classify realistic frames and realistic motions as before, while the generator will try to produce a realistic video such that the first frame is reconstructed well.

**Results:** We qualitatively show a few examples of our approach in Figure 5 using held-out testing videos. Although the extrapolations are rarely correct, they often have fairly plausible motions. The

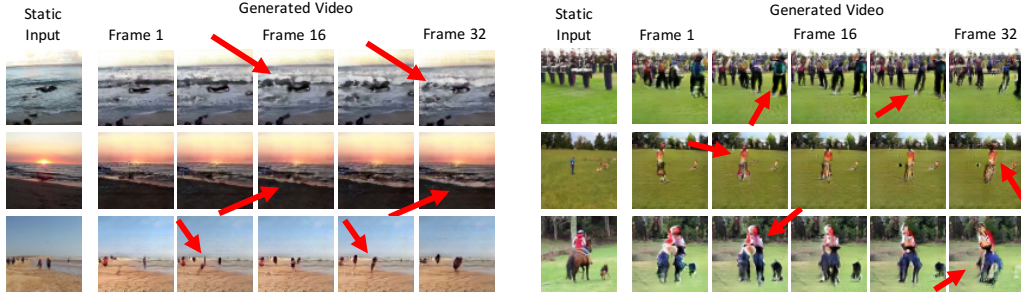

Figure 5: **Future Generation:** We show one application of generative video models where we predict videos given a single static image. The red arrows highlight regions of motion. Since this is an ambiguous task, our model usually does not generate the correct video, however the generation is often plausible. Please see `http://mit.edu/vondrick/tinyvideo` for animated movies.

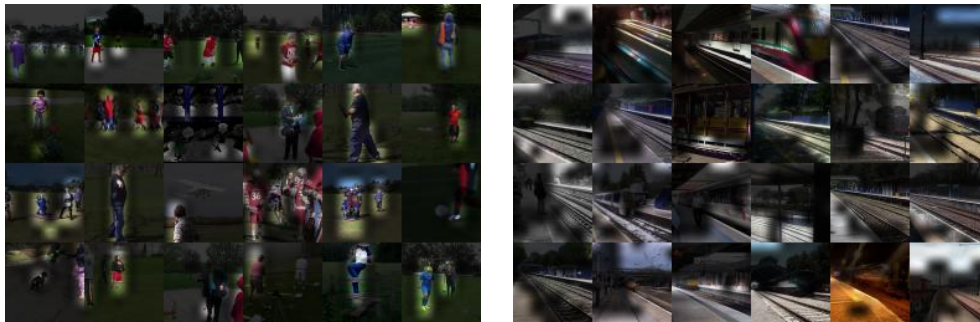

(a) hidden unit that fires on "person"  (b) hidden unit that fires on "train tracks"

Figure 6: **Visualizing Representation:** We visualize some hidden units in the encoder of the future generator, following the technique from [52]. We highlight regions of images that a particular convolutional hidden unit maximally activates on. While not at all units are semantic, some units activate on objects that are sources for motion, such as people and train tracks.

most common failure is that the generated video has a scene similar but not identical to the input image, such as by changing colors or dropping/hallucinating objects. The former could be solved by a color histogram normalization in post-processing (which we did not do for simplicity), while we suspect the latter will require building more powerful generative models. The generated videos are usually not the correct video, but we observe that often the motions are plausible. We are not aware of an existing approach that can directly generate multi-frame videos from a single static image. [33, 22] can generate video, but they require multiple input frames and empirically become blurry after extrapolating many frames. [43, 50] can predict optic flow from a single image, but they do not generate several frames of motion and may be susceptible to warping artifacts. We believe this experiment shows an important application of generative video models.

**Visualizing Representation:** Since generating the future requires understanding how objects move, the network may need learn to recognize some objects internally, even though it is not supervised to do so. Figure 6 visualizes some activations of hidden units in the third convolutional layer. While not at all units are semantic, some of the units tend to be selective for objects that are sources of motion, such as people or train tracks. These visualizations suggest that scaling up future generation might be a promising supervisory signal for object recognition and complementary to [27, 5, 46].

**Conclusion:** Understanding scene dynamics will be crucial for the next generation of computer vision systems. In this work, we explored how to learn some dynamics from large amounts of unlabeled video by capitalizing on adversarial learning methods. Since annotating dynamics is expensive, we believe learning from unlabeled data is a promising direction. While we are still a long way from fully harnessing the potential of unlabeled video, our experiments support that abundant unlabeled video can be lucrative for both learning to generate videos and learning visual representations.

**Acknowledgements:** We thank Yusuf Aytar for dataset discussions. We thank MIT TIG, especially Garrett Wollman, for troubleshooting issues on storing the 26 TB of video. We are grateful for the Torch7 community for answering many questions. NVidia donated GPUs used for this research. This work was supported by NSF grant #1524817 to AT, START program at UMBC to HP, and the Google PhD fellowship to CV.

## Footnotes

[1]See http://mit.edu/vondrick/tinyvideo for the animated videos.

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
