[Reviews · NeurIPS 2016]

Reviewer 1

Summary

This problem tackles the problem of generating short (with very few frames) tiny (64x64) videos. The approach uses a "deconvolutional" neural network that is split into two parts: the first predicts a foreground "motion", and a mask. The second predicts the motion of the background. The mask is then used to combine the two and obtain the desired generated video. In order to ensure that the videos generated are plausible, the networks are trained in an adversarial setup.

Qualitative Assessment

Overall this paper is very clearly laid out, and it is very easy to follow. Given that the authors are basing much of their method on existing methods for image generation, the novelty of the method lies in the way they adapted such methods to generate video. It is important to emphasize that I am not familiar with any other papers that attempt to do this (and the authors also didn't seem to be able to find other such papers). The problem with video, unlike images is that low frequencies are not only spanning space, but also time. Therefore, when generating video, typical methods will attempt to generate the temporal low frequencies first, resulting in very jarring outputs. The authors tackled this problem by explicitly decomposing the "background" from the "foreground". The background network's task is to generate the "low frequencies" while the foreground can focus (and will focus) on generating the more interesting parts (the high frequencies -- or motions of the "small" objects). In order to generate "plausible" images, the authors employ an adversarial critic network (or discriminator). In terms of the technical contents, outside the high level ideas, I have some questions for the authors: 1) why did you not use batch norm in the discriminator in the first layer? 2) how come your generator/discriminator don't use the same activation functions (i.e., ReLU vs Leaky ReLU)? In the final version of the paper, please attempt to describe the reasoning for these decisions, and possibly provide results showing a more consistent setup. Regarding the human eval results, I appreciate the honesty -- most of the videos don't seem "real", and some are rather jarring (especially the baby videos).

Confidence in this Review

2-Confident (read it all; understood it all reasonably well)


Reviewer 2

Summary

The paper describes a generative adversarial convolutional neural network for video (a block of 32 frames x 64x64 pixels). The architecture is divided into two streams one for static background and one for the moving foreground that are combined together using a mask that is also generated. The loss is a standard generative adversarial loss. The paper shows video generation experiments on four scene categories with qualitative results and quantitative user-study evaluating the realism of the generated videos. In addition, the paper uses trained representation on 5000 hours of Flickr videos for action classification demonstrating that the resulting representation serves as a better initialization for training action classification models.

Qualitative Assessment

Strengths: - The addressed problem of generative models of videos is interesting, timely and difficult. - Novelty: The proposed two stream architecture (static background / moving foreground + mask) for video generation is novel (albeit somewhat incremental over the previous static image generation GANs). Nevertheless, I like the proposed extension. - Experiments: The set of experiments is fairly comprehensive (generation, classification, user study). The results are encouraging, but but the visual quality of the generated results is quite poor and the action recognition results are much below the current state-of-the-art on the considered UCF dataset. Weaknesses: - The paper is somewhat incremental. The developed model is a fairly straighforward extension of the GAN for static images. - The generated videos have significant artifacts. Only some of the beach videos are kind of convincing. The action recognition performance is much below the current state-of-the-art on the UCF dataset, which uses more complex (deeper, also processing optic flow) architectures. Questions: - What is the size of the beach/golf course/train station/hospital datasets? - How do the video generation results from the network trained on 5000 hours of video look? Summary: While somewhat incremental, the paper seems to have enough novelty for a poster. The visual results encouraging but with many artifacts. The action classification results demonstrate benefits of the learnt representation compared with random weights but are significantly below state-of-the-art results on the considered dataset.

Confidence in this Review

2-Confident (read it all; understood it all reasonably well)


Reviewer 3

Summary

The paper to propose to use a two stream architecture to generate videos, where one stream generates the background frames and another one generates the foreground frames and a set of masks. By having two streams it allows to generate different motions for the background and the foreground. The model is trained using a VGAN approach and trained small video clips, the authors show promising results for video generation, video representation learning and for animating images.

Qualitative Assessment

After so much recent work in image generation using Generative Adversarial Networks it is nice to see a novel proposal for videos. The paper is well written with nice examples. It is not very clear how much of the performance is due to the data cleaning, for instead filtered videos with a pre-trained image model and camera stabilization. There are no experiments using unfiltered unlabeled videos for generation, or without stabilization. The results in action recognition in UC-101 are ok, but a comparison with a model using a pre-trained on image classification is missing. There is not evaluation of the animated images.

Confidence in this Review

2-Confident (read it all; understood it all reasonably well)


Reviewer 4

Summary

In this paper the authors use the generative adversarial network (GAN), specifically DC-GAN in order to generate tiny videos. They do so by using a two stream architecture. One generates the background still image and the other generates the foreground motion and mask. The mask comprises of a separate last layer of the foreground generator that uses sigmoidal activation. The authors in addition to generating videos from noise, show applications for action classification that is competitive and extrapolation of single frame image.

Qualitative Assessment

The proposed work is novel and the authors have provided sufficiently convincing evaluation of their method. The results for specific class videos look reasonable ( sometimes confusing human evaluators). Overall I think that the method and evaluation are good enough to justify acceptance of the paper. One point could be that the evaluation could perhaps consider an additional architecture where one initially generates a background frame that is animated in a one stream architecture. Right now the motion is separately generated from the background and is combined. In this case, the generation would be more coupled. In any case, this architecture is partially evaluated when the authors consider extrapolation of single frame images. A minor point is that few videos are available for actual evaluation. A thorough evaluation with more videos would be appreciated.

Confidence in this Review

2-Confident (read it all; understood it all reasonably well)


Reviewer 5

Summary

This paper proposed a tiny video generative model via GAN. The problem is challenging due to its huge dimensions. The authors propose a two-stream architecture by modeling foreground and background, respectively. The motion is modeled through 3D convolutional layers, which is a direct extension to the approaches of image generation through GAN [30, 23]. The results are visually pleasant in terms of appearance and motion. Moreover, this work also explores the task of unsupervised video representation learning through a well-trained discriminator.

Qualitative Assessment

The paper is novel in terms of exploring large-scale video generative model. It is generally well written and easy to understand. The main concern is that, the proposed model is too general and mid to high level structured contents of a video are not well modeled even they are trained in each scene category respectively. As a proposition of video generation problem, understanding the contents of both frames and sequences (e.g. objects, actions, etc.) is essential. However from the proposed architecture, the discriminator can hardly catch any mid to high level visual information since no such regularizations are enforced. The results show numerous unrealistic object shapes and irregular motions, which are far from realistic ones. This also applies to the results of animating images. Although the problem is interesting, some results leave much room for improvement before it can be presented (with convincing results) in NIPS.

Confidence in this Review

2-Confident (read it all; understood it all reasonably well)


Reviewer 6

Summary

The paper aims at generating short (32 frames) video segments with resolution 64x64. The proposed method relies on Generative Adversarial Network approach. There are 2 proposed architectures: straight-forward one and the one that generates static background and dynamic foreground separately. The training data is generated in a way that tries to "stabilize" images by compensating camera motion. This results in ignoring samples for which it can't be done reliably. There are 3 types of experiments: 1. video generation. Samples, generated from different architectures and self-implemented baseline, are compared with real videos by asking Amazon Mechanical Turk assessors to determine which of the two videos is real. Assessors prefer generated videos to real ones in 18% of the cases on average. 2. pre-training for action classification. The algorithm does not outperform current state-of-the-art Temporal Ordering. 3. generation of video conditioned on the first frame.

Qualitative Assessment

Experiments demonstrating large motion of large objects at larger resolution would be very important for understanding if the method works reasonably. Many scenes shown in the results either almost do not move (like hospital 1,3,5,6,7,8,9,11,12,13,14,15) or provide strange-looking movements (like oscillating face in hospital 2,10 and many train videos). Larger resolution would certainly be helpful as assessors would be able to better see what is actually happening in the video. At the current resolution it is very hard to see what exactly is happening to moving objects. Most of the sea videos just show flickering of the water, how natural it looks is also only possible to say for larger resolution.

Confidence in this Review

2-Confident (read it all; understood it all reasonably well)